Identification of immune related cells and crucial genes in the peripheral blood of ankylosing spondylitis by integrated bioinformatics analysis

Zheng Yang 1
Cai Bingbing 2
Ren Conglin 1
Xu Haipeng 1
Du Weibin 2
Wu Yijiang 1
Lin Fu 1
Zhang Helou 1
Quan Renfu 1 2 3 quanrenfu@126.com
1 Zhejiang Chinese Medical University , Hangzhou , China
2 Department of Orthopedics, Hangzhou Xiaoshan Hospital of Traditional Chinese Medicine , Hangzhou , China
3 Research Institute of Orthopedics, The Affiliated Jiangnan Hospital of Zhejiang Chinese Medical University , Hangzhou , China
Nakamura Tomoki
Electronic publication date: 2021 Sep 7
Publication date: 2021
Volume: 9
Electronic Location ID: e12125
Received 2021 Feb 17; Accepted 2021 Aug 17
Copyright: © 2021 Zheng et al.
Copyright year: 2021
Copyright holder: Zheng et al.
License: This is an open access article distributed under the terms of the Creative Commons Attribution License, which permits unrestricted use, distribution, reproduction and adaptation in any medium and for any purpose provided that it is properly attributed. For attribution, the original author(s), title, publication source (PeerJ) and either DOI or URL of the article must be cited.
License URL: https://creativecommons.org/licenses/by/4.0/

Keywords: Ankylosing spondylitis, Immune infiltration, Crucial genes, Bioinformatics analysis

Funding: National Natural Science Foundation of China 81904053 Zhejiang Provincial Science and Technology Program 2020KY797 Hangzhou City Science and Technology Program 20171226Y96 This study was supported by the National Natural Science Foundation of China (No. 81904053), the Zhejiang Provincial Science and Technology Program (No. 2020KY797), the Hangzhou City Science and Technology Program (No. 20171226Y96), and Zhejiang Chinese Medical University (No. ZYX2018008). The funders had no role in study design, data collection and analysis, decision to publish, or preparation of the manuscript.

==============================
Background

Ankylosing spondylitis (AS) is a progressive rheumatic disease and studies reveal that the immune system is critical for the pathogenesis of AS. In the present study, various bioinformatics analysis methods were comprehensively applied, designed to identify potential key genes and inflammation states of AS.

Methods

The transcriptome profiles of GSE25101 and GSE73754 obtained from the Gene Expression Omnibus (GEO) database were merged for subsequent analyses. The differentially expressed genes (DEGs) were identified using the Bioconductor package Limma and threshold values. Functional enrichment and pathway enrichment analyses were performed using the clusterProfiler package and Gene Set Enrichment Analysis (GSEA). Next, protein–protein interaction (PPI) network of the identified DEGs was constructed by the online database, the Search Tool for the Retrieval of Interacting Genes (STRING), visualization and analysis were performed through Cytoscape software. Subsequently, we applied CIBERSORT algorithm to identify subpopulation proportions of immune cells in peripheral blood samples. Finally, we validated the hub genes with the GSE18781 dataset. Samples were collected from patients to validate gene and protein expression using qRT-PCR and ELISA.

Results

A total of 334 DEGs were identified, including 182 upregulated and 152 downregulated DEGs, between AS patients and normal human controls, which were primarily involved in immune response, autophagy, and natural killer cell-mediated cytotoxicity. The most prominent module and candidate biomarkers were identified from the PPI network. Biomarkers were selected for validation and their expressions were significantly decreased in peripheral blood samples which was consistent with transcriptome sequencing results. Nine genes with AUC > 0.70 were considered to be AS hub genes for ROC curve analysis, including GZMA, GZMK, PRF1, GNLY, NKG7, KLRB1, KLRD1, IL2RB and CD247. Furthermore, CIBERSORT results suggest that AS contained a higher proportion of CD8+ T cells, naive CD4+ T cells, neutrophils, and lower levels of gamma delta T cells compared with the normal controls.

Conclusion

In this study, we identified DEGs combined with their closely related biological functions and propose that granule-associated proteins and immune infiltration maybe involved in the progression of ankylosing spondylitis. These validated hub genes may provide new perspectives for understanding the molecular mechanisms of ankylosing spondylitis.

Introduction

Ankylosing spondylitis (AS) is a chronic autoimmune inflammatory disease mainly involving the axial skeleton. The onset of this disease is predominantly seen in males before age 40 (Braun & Sieper, 2007; Chen et al., 2019; Smith, 2015). It is estimated that its prevalence in the US is 0.9% to 1.4% of the adult population, which is similar to rheumatoid arthritis (Reveille, Witter & Weisman, 2012). Its clinical manifestations and diagnostic points include chronic axial stiffness or inflammatory back pain, improvement with exercise, imaging data of the pelvis and spine, and serological examinations, especially for the HLA–B27 biomarker (Taurog, Chhabra & Colbert, 2016). AS onset is associated with a combination of hereditary risks, the intestinal microbiome, and the immune response. However, a number of studies have reported that human leukocyte antigen B27 (HLA-B27) is closely related to AS susceptibility. The genetic risk posed by HLA-B27 causes a predisposition toward AS and accounts for approximately 30% of cases, suggesting that other elements may also play a critical role in the development of this disease (Hu et al., 2020; Mohammadi et al., 2018).

Most studies have reported ankylosing spondylitis as an autoimmune disease marked by the activation of the immune system and subsequent chronic inflammation. These symptoms are confirmed by distinct inflammatory cell infiltration in the involved joints and by the therapeutic efficacy of NSAIDs and TNFα blockers (Duan et al., 2017; Vanaki et al., 2018). However, it is not fully understood how the immune system changes in AS and how these alterations meditate arthritis. Studying abnormal immune infiltrating cells may provide new strategies for treatment and diagnosis in light of the immune cell imbalance in AS.

Multiple gene expression profiling studies using microarray and bioinformatics analysis have been conducted to identify disease-associated biomarkers and pathways. These studies have attempted to highlight transcriptome differences between various phenotypes and stages of disease. The microarray technique in particular is a powerful tool for exploring gene regulation patterns and molecular mechanisms related to oncogenesis and progression of AS. However, the results of these studies are limited or inconsistent due to significant variability between different projects or small sample sizes. Integrated bioinformatics approaches have been applied in biological research to assist in the development of more sensitive and effective diagnostic and therapeutic strategies. We downloaded the transcriptome datasets for AS patients and healthy controls from the Gene Expression Omnibus (GEO) database to identify key genes and the associated biological processes. These results may improve our understanding of AS development and help create new diagnostic and immunoregulatory therapeutic targets for treatment.

Materials and Methods

Microarray data

We used the GEO database (http://www.ncbi.nlm.nih.gov/geo) to obtain gene expression profiles for ankylosing spondylitis. A total of 36 healthy and 68 diseased specimens were collected from the gene expression profiles of GSE25101 and GSE73754. Additionally, the mRNA profiles of 25 healthy and 18 diseased specimens in the GSE18781 dataset were used to detect whether discovered hub genes had good diagnostic value for AS.

Data normalization

The series matrix for each expression dataset was also obtained from the GEO database. R software (version 4.0.1; https://www.r-project.org/) and Bioconductor packages (http://www.bioconductor.org/) were used for the bioinformatics analysis. First, the “affy” package in R was used to conduct normalization and background correction. Next, probe expression data were converted into gene expression matrices according to the platform annotation file. The average expression value was used for further analysis for various probes corresponding to one gene. Lastly, the “sva” R package was applied to eliminate differences between the 68 AS patient and 36 healthy control samples (Leek et al., 2012).

Differentially expressed genes

In R the limma package used to screen out differentially expressed genes (DEGs) based on the comparison of expression data between AS samples and normal samples with the selection criteria of adjusted P < 0.05 and |logFC| > 0.2. We used the limma and pheatmap packages to create volcano plots and heatmaps for DEGs in RStudio (version:1.3.959) (Ritchie et al., 2015).

Functional enrichment analysis for DEGs

The Bioconductor package clusterProfiler (Yu et al., 2012) was used to evaluate DEGs’ Gene Ontology (GO) and Kyoto Encyclopedia of Genes and Genomes (KEGG) pathway enrichment analysis. We found significant biological processes and pathways. GO terms and KEGG pathways with significant enrichment were identified using an adjusted P of <0.05. Gene Set Enrichment Analysis (GSEA) is a computational method for analyzing statistically significant and concordant differences between two groups (e.g., AS and the control) using a priori defined gene set (Subramanian et al., 2005). The expression values and phenotype labels in the GSE25101 and GSE73754 datasets were used to conduct GSEA analysis according to AS status (AS vs. non-AS) with the KEGG gene sets as a reference. We set the cut-off criteria as nominal P value < 0.01 and false discovery rate (FDR) < 0.25. The results were shown using the “ggplot2” package in R.

PPI network construction and analysis

The STRING database (http://www.string-db.org/) was used to search for associations between known and predicted proteins. This database is commonly used to evaluate PPI information by calculating their combined score. Cytoscape 3.5.0 (version 3.7.2; https://cytoscape.org/) was used to construct and visualize the results from the PPI network; the criteria was a combined score of > 0.4 (Otasek et al., 2019). Cytoscape’s plugin MCODE, which can find the most highly connected cluster in a complex network, was used to extract the most significant module in the PPI networks. CytoHubba (Chin et al., 2014) was utilized to calculate the degree centrality of protein nodes to search for hub genes using the Maximal Clique Centrality (MCC) method. Results were ranked based on MCC scores. These applications were used to identify key modules and hub genes in the PPI network with default parameters.

Estimation of immune cell subtype distribution

The CIBERSORT algorithm (Fan et al., 2019) was used to transform the normalized gene expression value into the component of peripheral blood immune cells with the LM22 reference signature matrix set at 1,000 permutations. We retained the CIBERSORT output with P value of <0.05 for subsequent analysis and calculated the percentage of immune cell subtypes in all samples. We applied CIBERSORT to characterize the immune cells in AS and control samples. In R we used bar and violin plots to depict the infiltration levels of immune cells from different subpopulations between the two groups.

Validation of the hub genes by quantitative real-time PCR

Peripheral blood samples from six AS patients and six patients without AS were collected for qRT-PCR validation to confirm the results of our bioinformatics analysis. The study protocol was approved by the Ethics Committee of Xiaoshan Traditional Chinese Medical Hospital, and all patients signed the informed consent. Mononuclear cells from peripheral blood were isolated by density gradient separation using standard operating procedures (STEMCELL Technologies Inc., Vancouver, Canada). Four milliliters of peripheral blood were collected in heparin sodium tubes prior to treatment. The blood was diluted with an equal amount of Dulbecco’s phosphate buffered saline with 2% fetal bovine serum (STEMCELL Technologies Inc., Vancouver, Canada). Samples were then layered on top of lymphoprep (STEMCELL Technologies Inc., Vancouver, Canada) and centrifuged at 800 × g for 20 min at room temperature. The layer of mononuclear cells was washed twice with phosphate buffered saline. Total RNA from peripheral blood was extracted using a RNA-Quick Purification Kit (Yishan Biotechnology, Shanghai, China). The concentration and purity of total RNAs were measured with Nanodrop One (Thermo Scientific Inc., Waltham, MA, USA), and only RNA samples with Abs260 nm/Abs280 nm ratio > 1.8 were used. Total RNAs were reverse transcribed to cDNA using a RevertAid First Strand cDNA Synthesis Kit (Thermo Scientific Inc., Waltham, MA, USA), and qRT-PCR was conducted using the PowerUp™ SYBR™ Green Mix (Thermo Scientific Inc., Waltham, MA, USA) on the Applied Biosystems™ 7,500 (Thermo Scientific Inc., Waltham, MA, USA). We used GAPDH as an internal control. All primers used in this study are shown in Table 1. The relative mRNA expression was normalized and calculated using the comparative Ct method (2-ΔΔCt). The data were presented as the fold change of expression relative to normal tissues. One-way analysis of variance was conducted for comparison, and P < 0.05 indicated statistically significant differences.

Table 1 Primer and oligo sequences that were used in the study.

Primer name	Primer sequence, 5′–3′	
GZMK	Forward: GTGTTCTGATTGATCCACAGTG
Reverse: CATGTTTATTGAGTTTTGCGGC	
GNLY	Forward: CTACAGGACCTGTCTGACGATA
Reverse: CAGCATTGGAAACACTTCTCTG	
KLRD1	Forward: GTGAACAGAAAACTTGGAACGA
Reverse: ATAGATACTGGGAGAGTGCAGA	
GZMB	Forward: GAAAGTGCGAATCTGACTTACG
Reverse: TTGTTTCGTCCATAGGAGACAA	
PRF1	Forward: GCTATCGTTAGTGCTAGTGGAT
Reverse: ATCTGTCTGATGCGTATCCAAT	
IL2RB	Forward: CTGAGATCTCGCCACTAGAAG
Reverse: GGAAGAAGAAGTAACCCTGGTT	
KLRB1	Forward: AATTTGCCCTGAAACTTAGCTG
Reverse: GGATGTCACTGAAACACTCAAC	
NKG7	Forward: GATGTTCTGCCTGATTGCTTTG
Reverse: GGACAAGGACAAGAGAGATGG	
CD247	Forward: ATAACGAGCTCAATCTAGGACG
Reverse: CTGTACTGAGACCCTGGTAAAG	
GZMA	Forward: GCGAAGGTGACCTTAAACTTTT
Reverse: TGACTTCTCTCAGAGTATCGGA	
GAPDH	Forward: GGAGCGAGATCCCTCCAAAAT
Reverse: GGCTGTTGTCATACTTCTCATGG	

Validation of hub genes with the GEO database

Genes in the PPI network identified by the MCODE plug-in were selected as key candidate genes. We applied ROC analysis to evaluate the prediction efficacy of these candidate genes and calculated the area under curve (AUC) with RStudio. The genes with an AUC of >0.7 as well as a P-value of <0.05 were considered to be AS hub genes.

Enzyme‑linked immunosorbent assay

Serum was obtained from three AS patients and three patients without AS. The serum level of GZMA, GNLY, and PRF1 were analyzed with a Human Granulysin ELISA Kit (EK1280), Human Granzyme A ELISA Kit (EK1162), and Human PRF1 ELISA Kit (MM-0254H2) according to the manufacturers’ protocols. The absorbance was determined to be 450 nm by EnSpire (PerkinElmer, Waltham, MA, USA).

Results

Identification of DEGs

Raw data were read, corrected, and normalized. We obtained 334 DEGs including 182 upregulated and 152 downregulated genes after batch correction and normalization of the integrated dataset (GSE25101 and GSE73754) (Table S1). All DEGs were shown using a volcano plot (Fig. 1A) and the expression of the top 50-upregulated and downregulated genes was shown with a heatmap (Fig. 1B).

Figure 1 Visualization of differentially expressed genes (DEGs).

(A) DEGs screened by threshold (adjusted P value < 0.05 and |logFC| > 0.2) were presented by volcano plot. (B) Heatmap showed the expression of top 50 upregulated and downregulated genes ordered by adjusted P-value. Red indicated that the expression of genes was relatively upregulated, and the blue indicated the expression of genes was relatively downregulated.

GO and KEGG enrichment analysis of DEGs

Functional enrichment analyses were performed using the clusterProfiler package for further insight into the biological functions and pathways related to DEGs. For the GO analysis (Table 2 and Fig. 2A), DEGs were primarily enriched in I-kappaB kinase/NF-kappaB signaling, immune cell related processes, autophagy, process utilizing autophagic mechanism, positive regulation of cytokine production, and response to molecule of bacterial origin. KEGG pathway analysis showed that DEGs primarily participated in Phagosome, Leishmaniasis, Hematopoietic cell lineage, Natural killer cell-mediated cytotoxicity, Necroptosis, Apoptosis, Graft-vs.-host disease, Acute myeloid leukemia, and Th1 and Th2 cell differentiation (Table 3 and Fig. 2B).

Figure 2 Results of functional enrichment analysis.

(A) GO analysis results of DEGs, top 15 terms in BP category were listed. (B) The top nine pathways of KEGG analysis (according to the adjusted P value).

Table 2 GO analysis results of DEGs (top 15 terms of BP category were listed). ‘‘Count’’ means how many DEGs are involved.

ID	Description	p.adjust	Count	
GO:0043122	Regulation of I-kappaB kinase/NF-kappaB signaling	3.00E − 05	17	
GO:0007249	I-kappaB kinase/NF-kappaB signaling	3.00E − 05	18	
GO:0043123	Positive regulation of I-kappaB kinase/NF-kappaB signaling	3.00E − 05	15	
GO:0042119	Neutrophil activation	5.63E − 05	24	
GO:0043312	Neutrophil degranulation	9.15E − 05	23	
GO:0002228	Natural killer cell mediated immunity	9.15E − 05	9	
GO:0002283	Neutrophil activation involved in immune response	9.15E − 05	23	
GO:0002446	Neutrophil mediated immunity	0.000119	23	
GO:0001819	Positive regulation of cytokine production	0.000121	22	
GO:0031331	Positive regulation of cellular catabolic process	0.000157	19	
GO:0009896	Positive regulation of catabolic process	0.000391	20	
GO:0051091	Positive regulation of DNA-binding transcription factor activity	0.000734	15	
GO:0006914	Autophagy	0.000883	21	
GO:0061919	Process utilizing autophagic mechanism	0.000883	21	
GO:0002237	Response to molecule of bacterial origin	0.000917	17	

Table 3 KEGG analysis results of DEGs (top ten pathways were listed).

‘‘Count’’ means how many DEGs are involved.

ID	Description	p.adjust	Count	
hsa04145	Phagosome	0.000184	14	
hsa05140	Leishmaniasis	0.001378	9	
hsa04640	Hematopoietic cell lineage	0.006969	9	
hsa04650	Natural killer cell mediated cytotoxicity	0.009257	10	
hsa04217	Necroptosis	0.034964	10	
hsa04210	Apoptosis	0.034964	9	
hsa05332	Graft-vs.-host disease	0.034964	5	
hsa05221	Acute myeloid leukemia	0.041801	6	
hsa04658	Th1 and Th2 cell differentiation	0.041801	7	
hsa04380	Osteoclast differentiation	0.050315	8	

GSEA analysis

We used GSEA to perform a pathway enrichment analysis, defined by KEGG gene sets, to further explore the biological function of DEGs. According to the pre-set threshold of analysis results, complement and coagulation cascades, leukocyte transendothelial migration, osteoclast differentiation, and autophagy-animal gene sets were significantly upregulated in AS samples. Results are detailed in Fig. 3.

Figure 3 GSEA pathway enrichment analysis.

Pathway with FDR < 0.05 were considered to be significant.

PPI network construction and analysis

To further explore the relationships between DEGs at the protein level, the PPI network of DEGs was constructed by online STRING database with the predefined criteria, and visualized using Cytoscape. Upregulated and downregulated nodes (DEGs) were labeled with red and blue, respectively (Fig. 4A). The diameters of the nodes indicated their degree of connectivity. The most significant module with the highest score (9.000) detected by MCODE analysis is shown in Fig. 4B. It consists of 23 genes and 99 edges. The top ten critical genes were selected by calculating MCC scores using the cytoHubba plugin; these genes are presented in Fig. 4C. All critical genes were downregulated in AS samples compared to control samples. The ten candidate hub nodes were all contained in the module mentioned above, which implied that the module may be a good representation of critical biological characteristics. The ten nodes were thus defined as the key nodes in the PPI network. The KEGG of these DEGs was analyzed using the clusterProfiler package. Our results showed that these genes were mainly enriched in the functions of natural killer cell mediated cytotoxicity, graft-versus-host disease, Th1 and Th2 cell differentiation, and Th17 cell differentiation (Table 4).

Figure 4 PPI network construction and module analysis.

(A) The PPI network of DEGs was constructed in Cytoscape, Nodes represent proteins and edges represent interactions between two proteins. (B) The most significant module was obtained by MCODE plug-in. Red nodes represent upregulated DEGs and blue nodes represent downregulated DEGs. The diameters of nodes were positively correlated with their connectivity degree. (C) Hub genes selected by calculating the MCC scores of cytoHubba plugin.

Table 4 Significantly enriched KEGG pathways for hub genes.

‘‘Count’’ means how many hub genes are involved.

ID	Description	p.adjust	Count	
hsa04650	Natural killer cell mediated cytotoxicity	5.58E − 05	4	
hsa05332	Graft-vs.-host disease	5.64E − 05	3	
hsa05330	Allograft rejection	0.003575	2	
hsa04940	Type I diabetes mellitus	0.003575	2	
hsa05320	Autoimmune thyroid disease	0.004346	2	
hsa04658	Th1 and Th2 cell differentiation	0.010825	2	
hsa04659	Th17 cell differentiation	0.012492	2	
hsa04210	Apoptosis	0.017483	2	
hsa05202	Transcriptional misregulation in cancer	0.030326	2	

Distribution pattern of immune cell subtype

The overall distribution of different immune subsets in all samples was shown in a histogram (Fig. 5A). Different colors and heights represent various types and percentages of immune cells in the sample, and the sum of the proportion of various immune cells is 1. CD8+ T cells, naive CD4+ T cells, gamma delta T cells, and neutrophils were the primary infiltrating cells. There are individual differences in the proportion of immune cells between the two groups. Clustering analysis of infiltrating immune cells in AS and control samples is a vital for identifying immunopathogenesis (Fig. 5B). The different immune cell subsets were weakly-to-moderately correlated. The correlation of NK cells and neutrophils was 0.5, the correlation between neutrophils and CD8+ T cells was 0.66 (Fig. 5C). Compared with healthy controls, the proportion of CD8+ T cells (P = 0.007), naive CD4+ T cells (P = 0.041), and neutrophils (P = 0.007) was higher in AS samples, while the fraction of gamma delta T cells (P = 0.012) was relatively lower (Fig. 5D).

Figure 5 The profiles of immune cell subtype distribution pattern in GSE25101 and GSE73754 cohort.

(A) The bar plot visualizing the relative percent of 22 immune cell in each sample. (B) Heatmap of the 22 immune cell proportions in each sample. (C) Correlation heatmap of all 22 immune cells. (D) Violin plot of all 22 immune cells differentially infiltrated fraction.

Validation of candidate biomarkers

The qRT-PCR results showed a significant downregulation in the expression of ten hub genes (GZMK, GNLY, KLRD1, GZMB, PRF1, IL2RB, KLRB1, NKG7, CD247, and GZMA) obtained from cytoHubba analysis. These results were consistent with the results of the microarray hybridization, suggesting that the results were convincing (Fig. 6). ELISA analysis suggested that protein expression levels of GZMA, GNLY, and PRF1 in serum were significantly decreased in the AS group compared to the control group (Fig. 7).

Figure 6 RT-PCR validation of the hub gene between AS and normal controls.

(A) CD247, (B) GNLY, (C) GZMA, (D) GZMB, (E) GZMK, (F) IL2RB, (G) KLRB1, (H) KLRD1, (I) NKG7 and (J) PRF1. (*P < 0.01).

Figure 7 ELISA validation of the protein expression.

ELISA analysis of GZMA, GNLY and PRF1 expression in serum between AS and normal controls. (*P < 0.05).

Receiver operating characteristic (ROC) curve analysis

The validation dataset (GSE18781) was obtained from the GEO database. We performed ROC curve analysis using RStudio to further verify the reliability of these 10 candidate genes in patients’ peripheral blood. Nine of the ten genes (GZMA, GZMK, NKG7, PRF1, GNLY, KLRD1, KLRB1, IL2RB, CD247) with an AUC of more than 0.70 were considered hub genes, indicating that they may be able to diagnose AS patients with excellent specificity and sensitivity (Fig. 8).

Figure 8 Validation of candidate hub genes by ROC curve analysis.

Among the ten genes screened out by cytoHubba plug-in, nine genes with AUC more than 0.70 were considered as hub genes of AS.

Discussion

AS is a chronic inflammatory rheumatic disease characterized by two major features: chronic inflammation and progressive ankylosis in the axial skeleton. This disease leads to a clinical picture that includes inflammatory back pain, asymmetrical peripheral oligoarthritis, chronic inflammatory bowel disease (IBD), progressive stiffness, and loss of spinal mobility (Appel et al., 2006; Blair, 2019; Braun & Sieper, 2007). The interaction between host genetics, the intestinal microbiome, and the immune response is strongly related to AS pathogenesis (Qiao et al., 2019; Wang et al., 2016; Yang et al., 2016). Previous bioinformatics research found that some immune-associated pathways like T-cell receptor signaling pathway, natural killer cell mediated cytotoxicity were significantly enriched in AS patients (Chen et al., 2012; Xu et al., 2019). And one bioinformatics study revealed that some TNF and interleukin (IL) related factors were detected as DEGs, such as IL2RB, IL17RB and IL17RD, which might be associated with in the progression of AS (Zhao et al., 2015). These studies have demonstrated the importance of immune cells in determining the pathogenesis of AS. Given the progressive and often disabling nature of AS, it is important to identify the novel molecular targets and potential mechanisms of AS to provide underlying biomarkers or therapeutic approaches. We screened out 334 DEGs, including 182 upregulated and 152 downregulated genes, by comparing the differences in gene expression profiles of peripheral blood in AS patients and healthy controls. Functional enrichment analysis was performed to further explore the regulatory roles of DEGs in AS. We constructed a PPI network containing DEGs in the STRING database. Cytoscape software was used to visualize interactions between DEGs. The MCODE plug-in and Cytohubba were used to screen the most significant module and hub genes in the PPI network.

The results of enrichment analysis showed that the module and DEGs were significantly correlated with immune-related functions and inflammatory signaling, such as the activation and regulation of I-kappaB kinase/NF-kappaB signaling, neutrophil activation, Th1 and Th2 cell differentiation, and Th17 cell differentiation. T helper (Th) cells are divided into Th1 and Th2 subsets according to their cytokine production profiles (Abbas, Murphy & Sher, 1996). Previous studies found that Th1 and Th2 chemoattractants played a cooperative role in the development of AS. Th1 and Th2 chemokine levels decreased under etanercept, a TNF-α blocker, to improve the activity and functional capacity of AS patients (Ergin et al., 2011; Wang et al., 2016; Zhang et al., 2018). Animal experiments indicated that Micheliolide, a sesquiterpene lactone involved in alleviating the inflammatory response, maintained the balance of Th1/Th2 by regulating NF-κB signaling (Tian, Yao & Chen, 2020). Th17 cells were shown to be involved in AS (Xueyi et al., 2013). T-helper 17 (Th17) cells are a subset of CD4+ T cells and were found to be related to the pathogenesis of autoimmune diseases and inflammatory diseases such as AS, rheumatoid arthritis (RA), and inflammatory bowel disease (IBD) (Hammitzsch et al., 2018). Th17 cells produce proinflammatory cytokine and interleukin-17 (IL-17), and their differentiation is regulated by the presence of IL-23 (Hajialilo et al., 2019). In addition, antibodies targeting the IL-23/IL-17 axis have demonstrated efficacy; a genome-wide association study of AS found that IL-23 and IL-1 cytokine pathways play a role in disease susceptibility (Reveille et al., 2010). This effect plays a key role for T helper cell type 17 (T17) responses in the pathogenesis of AS (Baeten et al., 2015; Karaderi et al., 2009). These findings imply that T helper cell responses are instrumental for the pathogenesis of AS and deserve more attention.

We analyzed the proportion of blood immune cell subsets using CIBERSORT, a transcriptome deconvolution algorithm, to better understand the role of leukocyte infiltration and the inflammatory response in the pathogenesis and development of AS. Of the 22 major immune cell types, we found significant differences in the composition of immune cells between AS and healthy controls. These differences include an increase in the proportion of CD8+ T cells, naive CD4+ T cells, neutrophils, and the lower portion of Gamma Delta T cells. These differences were also significantly associated with AS. Our results were consistent with previously reported variations in the abundancies of various cells in AS (Shamji, Bafaquh & Tsai, 2008; Zhang et al., 2009).

GO and GSEA enrichment analysis indicated that autophagy may play a vital role in the pathological development of AS. Autophagy is a cellular lysosomal degradation pathway to remove pathogens and dead cells. Autophagy also inhibits inflammasome activation and secretion that is important for cellular homeostasis (Ma et al., 2019; Zhai et al., 2018). It is well known that autophagy plays a potentially key role in immune cells’ induction and regulation of inflammatory responses. Therefore, dysfunction in this process may lead to inflammation, such as multiple sclerosis and rheumatoid arthritis (Sutton et al., 2006). Ciccia et al. (2014) found that HLA-B27 misfolding occurs in AS patients’ guts and is accompanied by autophagy activation rather than an unfolded protein response. A recent study illustrated the lower expression of autophagy-related genes and defective autophagy activity in AS patients’ peripheral blood (Park et al., 2017). This research suggested that autophagy plays a role in AS pathogenesis.

Finally, we validated these crucial genes in PPI network by performing qRT-PCR and ELISA assay. As a result, ten genes with P-value < 0.01 were successfully validated for their low expression in AS patients, including granzyme family (GZMA, GZMB and GZMK), perforin one (PRF1), granule proteins (GNLY and NKG7), killer cell lectin like receptor (KLRB1 and KLRD1), and T-cell signaling genes (IL2RB and CD247). And 9 genes validated by ROC curve analysis with P-value < 0.05 as well as AUC > 0.70 showed potential diagnostic value for AS, and thus were considered as hub genes of AS, including GZMA, GZMK, PRF1, GNLY, NKG7, KLRB1, KLRD1, IL2RB and CD247.

Notably, many crucial genes expressed at very low levels in AS patients were linked to the lymphoid lineage, especially cytotoxic lymphocytes including cytotoxic T lymphocytes (CTLs) and natural killer (NK) cells. Cytotoxic T lymphocytes (CTLs) and natural killer (NK) cells are characterized by potent toxins secretion, including granule proteins, perforins, and granulysins stored in the secretory granules (SGs) of CTLs and NK cells (Anthony et al., 2010). They employ SGs to cleave and activate effector molecules within the target cell through granule exocytosis (Voskoboinik, Whisstock & Trapani, 2015).

The KLRD1 and KLRB1 protein, expressed by NK cell, are classified as a type II membrane protein and may be involved in the regulation of NK cell function. A recent bioinformatics found KLRD1 and CD247 DEGs were significantly enriched in natural killer cell mediated cytotoxicity, regulation of immune response, which was consistent with our findings (Fan et al., 2019). IL2RB, an interleukin two receptor, is involved in T cell-mediated immune responses, which have been proved related to AS in previous study and a recent clinical study found significant associations with the presence of peripheral arthritis at AS onset in SNPs of IL2RB gene (Polo et al., 2019; Zhu, Tang & Cao, 2013).

The granzyme family is a class of homologous serine proteases, mainly expressed by cytotoxic T lymphocytes (CTLs) and natural killer (NK) cells (Fehniger et al., 2007; Pardo et al., 2002; Shah et al., 2011). To date, five various human granzymes (GZMA, GZMB, GZMH, GZMK, and GZMM) have already been identified according to the primary substrate specificity (Van Daalen, Reijneveld & Bovenschen, 2020). In addition, perforin and granzyme are tightly packed in the core of SGs and jointly mediate target cell apoptosis once SGs conjugated to a target cell (Trapani et al., 2013). Granulysin, encoded by GNLY, is also known as NKG5 and is similar to granzymes. It presents in SGs, diffuses into target cells through perforin pores and can induce the recruitment and activation of antigen-presenting cells (APCs), facilitating the generation of an immune response (Clayberger & Krensky, 2003; Tewary et al., 2010). Increasing evidence indicates that the abnormal expression of GZM family, granule proteins (NKG5 and NKG7), and perforin one (PRF1) is involved in a number of autoimmune diseases, including ankylosing spondylitis, rheumatoid arthritis (RA), hypersensitive pneumonitis, celiac disease, and scleroderma (Anthony et al., 2010; Prager et al., 2019). A previous study indicated the importance of PRF1 in immune response and immune regulation related functions in AS and highlighted its significance of common markers for rheumatic diseases (Wang et al., 2015). In addition, a recent cohort study confirmed that GZMA and PRF1 in AS and RA patients’ peripheral blood were significantly downregulated compared to the controls, especially for male AS patients. On the other hand, GZMA and GZMB in the synovial fluid (SF) of AS patients was significantly higher than in the serum, while PRF1 expression in SF was further downregulated (Gracey et al., 2020). Additional research should be conducted to address the potential roles of granule proteins secreted by SGs in the pathogenesis of ankylosing spondylitis.

Conclusions

In summary, we applied a comprehensive bioinformatics method to show that immune cell infiltration and granule-associated proteins were important in the pathogenesis of ankylosing spondylitis. The hub genes we identified may serve as potential biomarkers for AS. However, further experiments are required to support our conclusions.

Supplemental Information

Supplemental Information 1 The list of DEGs.

Click here for additional data file.

Supplemental Information 2 All code of R and Perl.

Click here for additional data file.

Supplemental Information 3 Raw Data.

Click here for additional data file.

Supplemental Information 4 Raw data of ELISA.

Click here for additional data file.

Supplemental Information 5 The baseline data of 9 AS patients.

Click here for additional data file.

Supplemental Information 6 GSE73754 series matrix.

Click here for additional data file.

Supplemental Information 7 GSE25101 series matrix.

Click here for additional data file.

Supplemental Information 8 GSE18781 series matrix.

Click here for additional data file.

Supplemental Information 9 Demographic analysis data of GSE datasets.

Click here for additional data file.

Supplemental Information 10 All significant GO pathways enriched by DEGs.

Click here for additional data file.

Supplemental Information 11 All significant KEGG pathways enriched by DEGs.

Click here for additional data file.

Supplemental Information 12 Heat map of immune infiltrating cells.

Based on CIBERSORT algorithm, the gene expression matrix of each sample is converted into the proportion matrix of immune cells in the sample. Dark red and dark blue represent the corresponding high proportion and low proportion of immune cells respectively.

Click here for additional data file.

Additional Information and Declarations

Competing Interests

Author Contributions

Human Ethics

Data Availability

The authors declare that they have no competing interests.

Yang Zheng conceived and designed the experiments, performed the experiments, analyzed the data, prepared figures and/or tables, authored or reviewed drafts of the paper, and approved the final draft.

Bingbing Cai performed the experiments, prepared figures and/or tables, and approved the final draft.

Conglin Ren analyzed the data, prepared figures and/or tables, and approved the final draft.

Haipeng Xu analyzed the data, prepared figures and/or tables, and approved the final draft.

Weibin Du performed the experiments, prepared figures and/or tables, and approved the final draft.

Yijiang Wu analyzed the data, authored or reviewed drafts of the paper, and approved the final draft.

Fu Lin analyzed the data, prepared figures and/or tables, and approved the final draft.

Helou Zhang performed the experiments, prepared figures and/or tables, and approved the final draft.

Renfu Quan conceived and designed the experiments, authored or reviewed drafts of the paper, and approved the final draft.

The following information was supplied relating to ethical approvals (i.e., approving body and any reference numbers):

The Ethics Committee of Xiaoshan Traditional Chinese Medical Hospital approved the study.

The following information was supplied regarding data availability:

The raw data are available in the Supplemental Files, including the list of differential gene expression, Perl and R language code, and the data of RT-PCR.

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
