# Peer review of "Identification of immune related cells and crucial genes in the peripheral blood of ankylosing spondylitis by integrated bioinformatics analysis"

_PeerJ, doi:10.7717/peerj.12125_

## Round 0.1 · original submission · Major Revisions

Now, the comments to the authors are ready. Please address them.

Reviewer 1 ·

Basic reporting

The article is clearly written but with some mistakes in typewriting, e.g., line 117, 203 and 207, et.al., which should be carefully checked and corrected.

Experimental design

The authors should state why the two datasets GSE25101 and GSE73754 were chosen for analysis, with other datasets from AS patients’ peripheral blood available, such as GSE18781. What’s the criteria of including and excluding datasets? And how these criteria would affect the results?
What’s the disease activity of the 6 AS patients and what medications they were receiving at the time sample collected?

Validity of the findings

no comment

Reviewer 2 ·

Basic reporting

I think this is an epoch-making treatise that comprehensively analyzes the genetic pathology of ankylosing spondylitis, which has not been completely elucidated yet.
However, I have to say that the pursuit of research is insufficient to conclude from this treatise.

Experimental design

no comment

Validity of the findings

no comment

Additional comments

The detailed patient background of ankylosing spondylitis is not fully considered.
For example, there is no mention of the duration of illness, treatment details, involvement of underlying diseases other than ankylosing spondylitis, etc., and their effects cannot be ignored.
Consideration should be given to the patient background.
If it is difficult to present a list of patient backgrounds, it is necessary to consider picking up AS patients excluding major diseases such as diabetes, collagen disease, and cancer. Is such an operation done?

Further verification is needed to determine whether this gene expression is completely linked to the pathogenic factors and pathophysiology of ankylosing spondylitis.

The authors are only conducting gene analysis, but do you confirm the protein expression level? If you have any data, please show it.
Also, are there any plans for in vitro cell-based research, in vivo animal experiments, confirmation with knockout mice, etc.? I think it is desirable to describe in the consideration part that the verification is necessary.

Also, the expression "useful for treatment" in the discussion department and conclusions is an overstatement. In other words.
As an additional study, it is necessary to analyze peripheral blood leukocytes in patients whose pathology can be controlled by intervention and those who do not. If you have data, please consider adding it.

Reviewer 3 ·

Basic reporting

There are a lot of grammatical errors, typos, or ambiguous parts in this manuscript, for examples:
- ... other elements may also play an critical role ...
- the comprehensive analysis may shed light on further insights regarding AS development and facilitate to provide new diagnostic and immunoregulatory therapeutic targets for AS.
- ... the GSE25101 and GSE73754 dataset ...
- ... with reference of KEGG gene sets ...
- the cut-off criteria as nominal nominal p value ...
- ...
Therefore, the authors have to re-check and revise carefully. It is better to be checked by a native speaker or an English editing service.

More literature reviews on bioinformatics-based ankylosing spondylitis studies should be added.

The part from line 70 to 83 should be in section "Methodology" or "Result".

There are some inconsistency in font format.

Experimental design

- A critical concern is the use of a very small dataset (only 36 normal and 68 diseased specimens). This number cannot be considered as enough as a comprehensive analysis.
- Methodology should be explained in more detail.
- GO database or analysis has been used in previous studies i.e., PMID: 31277574 and PMID: 31921391. Therefore, the authors are suggested to refer to more works in this description.
- Why did the structure of "Methodology" and "Results" look similarly? At least the authors should present the "Results" section in a different way, not list alike the method.

Validity of the findings

- The authors should compare the predictive performance with the previous studies on the same problem/data.
- The authors should have some validation data.
- What is the significant level of KEGG analysis?

Additional comments

No comment.

Reviewer 4 ·

Basic reporting

1. Citations for cell-specific expression of Granzyme expression in CTLs and NK cells to ascertain the RT-PCR findings are not contributed from other mononuclear cells.

2. Should also address what was missing in the original report of the GEO dataset generation and how this analysis generates different insights.

Experimental design

1. A detailed distinction in age, sex, education, and comorbidities of the participants in the GEO datasets should be factored into a demographic analysis to effectually address if any other factor could have affected the DEG comparison.

2. Brief description of the GEO datasets referenced - cells used, isolation procedure, storage conditions, RNA isolation procedure, and microarray analysis to directly compare the results with the RT-PCR analysis.

3. The above comment also applies to the RT-PCR analysis.

4. Detailed methods of peripheral blood analysis - such as a citation for the procedure of mononuclear cell isolation and methods used.

Validity of the findings

no comment

---

## Round 0.2 · Major Revisions

Please see the comments and make the revision for improving the manuscript.

Reviewer 1 ·

Basic reporting

After revising, the paper is much more readable, but still needs to be re-checked carefully throughout, such as “Figure 7” in line 237. The ABSTRACT section should also be revised, especially the part of Results, to make it stand alone without the full paper.

Experimental design

no comment

Validity of the findings

no comment

Reviewer 3 ·

Basic reporting

No comment.

Experimental design

No comment.

Validity of the findings

No comment.

Additional comments

Thanks for addressing my previous comments. However, in my opinion, some comments were not addressed well as follows:
- More literature reviews (related works) on bioinformatics-based ankylosing spondylitis studies should be added.
- The authors should compare the predictive performance with the previous studies on the same problem/data.
- The authors should have some validation data (the data with the same characteristics, not validation the protein expression)

Reviewer 4 ·

Basic reporting

No comment

Experimental design

No comment

Validity of the findings

No comment

---

## Round 0.3 · accepted · Accept

The manuscript is now ready for the publication.

Reviewer 3 ·

Basic reporting

No comment.

Experimental design

No comment.

Validity of the findings

No comment.

Additional comments

No comment.